# Establishing the Clinical Utility of ctDNA Analysis for Diagnosis, Prognosis, and Treatment Monitoring of Retinoblastoma: The Aqueous Humor Liquid Biopsy

**DOI:** 10.3390/cancers13061282

**Published:** 2021-03-13

**Authors:** Liya Xu, Mary E. Kim, Ashley Polski, Rishvanth K. Prabakar, Lishuang Shen, Chen-Ching Peng, Mark W. Reid, Patricia Chévez-Barrios, Jonathan W. Kim, Rachana Shah, Rima Jubran, Peter Kuhn, David Cobrinik, Jaclyn A. Biegel, Xiaowu Gai, James Hicks, Jesse L. Berry

**Affiliations:** 1The Vision Center at Children’s Hospital Los Angeles, Los Angeles, CA 90027, USA; lixu@chla.usc.edu (L.X.); maryekim@usc.edu (M.E.K.); ashley.polski@usc.edu (A.P.); ppeng@chla.usc.edu (C.-C.P.); mreid@chla.usc.edu (M.W.R.); jonkim@chla.usc.edu (J.W.K.); cobrinik@usc.edu (D.C.); 2Department of Biological Sciences, Dornsife College of Letters, Arts, and Sciences, University of Southern California, Los Angeles, CA 90007, USA; pkuhn@usc.edu (P.K.); james.hicks@usc.edu (J.H.); 3USC Roski Eye Institute, Keck School of Medicine of the University of Southern California, Los Angeles, CA 90033, USA; 4Department of Molecular and Computational Biology, University of Southern California, Los Angeles, CA 90007, USA; kaliappa@usc.edu; 5Center for Personalized Medicine, Department of Pathology and Laboratory Medicine, Children’s Hospital Los Angeles, Los Angeles, CA 90027, USA; lishen@chla.usc.edu (L.S.); jbiegel@chla.usc.edu (J.A.B.); xgai@chla.usc.edu (X.G.); 6Ophthalmic Pathology, Department of Pathology and Genomic Medicine, Houston Methodist Hospital, Houston, TX 77030, USA; pchevez-barrios@HoustonMethodist.org; 7Cancer and Blood Disease Institute at Children’s Hospital Los Angeles, Los Angeles, CA 90027, USA; rachana@usc.edu (R.S.); rjubran@chla.usc.edu (R.J.); 8Norris Comprehensive Cancer Center, Keck School of Medicine, University of Southern California, Los Angeles, CA 90033, USA; 9Department of Aerospace and Mechanical Engineering, Viterbi School of Engineering, University of Southern California, Los Angeles, CA 90007, USA; 10Department of Biomedical Engineering, Viterbi School of Engineering, University of Southern California, Los Angeles, CA 90007, USA; 11Department of Biochemistry and Molecular Medicine, Keck School of Medicine, University of Southern California, Los Angeles, CA 90033, USA; 12The Saban Research Institute, Children’s Hospital Los Angeles, Los Angeles, CA 90027, USA; 13Department of Pathology and Laboratory Medicine, Keck School of Medicine of USC, Los Angeles, CA 90033, USA

**Keywords:** retinoblastoma, aqueous humor, liquid biopsy, ctDNA, SCNA, precision oncology

## Abstract

**Simple Summary:**

Due to prohibition of direct tumor biopsy for patients with retinoblastoma, the prospect of a liquid biopsy for the identification of tumor derived biomarkers for this cancer is enticing. The aqueous humor (AH) is a rich source of eye-specific tumoral genomic information. This is the first prospective study wherein we demonstrate that molecular profiling of the AH at diagnosis and longitudinally throughout therapy has clinical utility for diagnosis, prognosis, and monitoring of treatment response. Tumoral genomic information was detected in 100% of diagnostic aqueous humor samples, including single nucleotide variants in the *RB1* tumor suppressor gene and large-scale somatic chromosomal alterations. All eyes that failed therapy and required enucleation had poor prognostic biomarkers for ocular salvage present in the aqueous humor *at time of diagnosis*. This highlights the potential of the AH liquid biopsy for direct clinical applications to precision oncology to direct genome-specific, personalized treatment for retinoblastoma patients.

**Abstract:**

Because direct tumor biopsy is prohibited for retinoblastoma (RB), eye-specific molecular biomarkers are not used in clinical practice for RB. Recently, we demonstrated that the aqueous humor (AH) is a rich liquid biopsy source of cell-free tumor DNA. Herein, we detail clinically-relevant molecular biomarkers from the first year of prospective validation data. Seven eyes from 6 RB patients who had AH sampled at diagnosis and throughout therapy with ≥12 months of follow-up were included. Cell-free DNA (cfDNA) from each sample was isolated and sequenced to assess genome-wide somatic copy number alterations (SCNAs), followed by targeted resequencing for pathogenic variants using a *RB1* and *MYCN* custom hybridization panel. Tumoral genomic information was detected in 100% of diagnostic AH samples. Of the seven diagnostic AH samples, 5/7 were positive for RB SCNAs. Mutational analysis identified *RB1* variants in 5/7 AH samples, including the 2 samples in which no SCNAs were detected. Two eyes failed therapy and required enucleation; both had poor prognostic biomarkers (chromosome 6p gain or *MYCN* amplification) present in the AH at the time of diagnosis. In the context of previously established pre-analytical, analytical, and clinical validity, this provides evidence for larger, prospective studies to further establish the clinical utility of the AH liquid biopsy and its applications to precision oncology for RB.

## 1. Introduction

Retinoblastoma (RB) is a primary intraocular cancer that forms in one or both eyes of infants and toddlers. The overall survival for these pediatric patients is high in the United States [1]; however, ocular survival is far less. Approximately half of eyes with advanced disease require surgical removal to prevent extraocular extension of disease [2]. While RB is quite rare, it has long been the prototypical cancer. Fundamental principles in cancer biology were discovered through the investigation of this orphan disease, including Knudson’s two-hit hypothesis for the *RB1* tumor suppressor gene [3] and the role of highly-recurrent somatic copy number alterations (SCNAs) in tumorigenesis [4,5,6,7,8]. Despite this knowledge, there have been no clinical applications of RB genomics, no eye-specific molecular biomarkers, nor any attempts at precision oncology management for this cancer because direct biopsy of RB tumors is strictly prohibited [9,10,11]. 

In 2017 this paradigm shifted when we demonstrated that the aqueous humor (AH), the clear liquid in front of the eye, is a high-yield, content rich source of tumor-derived cell-free DNA (cfDNA) that can be safely extracted. Unlike serum, AH is eye-specific and can be analyzed from each eye separately in children with bilateral intraocular disease. Thus, it is an ideal liquid biopsy platform for RB [12,13,14]. Multiple subsequent studies have connected RB genomics to clinically relevant outcomes, notably the ability to cure and save the eye with standard therapies [12,13,15,16,17,18,19]. Specifically, we demonstrated that chromosome 6p gain is associated with a nearly 10-fold increased odds of enucleation (surgical removal of the eye) [13,16], and that dynamics in cfDNA tumor fraction are closely associated with response to therapy [19]. We also published a comprehensive workflow for the AH liquid biopsy that included simultaneous whole-genome SCNA analysis, *RB1* pathogenic variant detection, and tumor fraction estimation from a single 100µL sample of AH from RB eyes [17]. 

Following technical and clinical validation, the AH is now established as an enriched liquid biopsy source of tumor-derived molecular information with potential clinical utility for the management of children with RB. An important distinction is that all previous studies of the AH liquid biopsy platform have focused on the safety and clinical research applications from eyes actively undergoing therapy, with retrospective analysis [12,13,14,15,16,17,18,19]. An accepted protocol for obtaining AH at the time of diagnosis (i.e., prior to surgical or chemotherapeutic treatment) has not yet been developed. This initial step in prospective validation of previous findings could only be initiated once we established analytical and clinical validation, with prolonged follow-up [12,13,15,16,17,18,19]. Herein, we describe the initial series of treatment-naive RB patients from whom AH was taken as part of their diagnostic evaluation under an approved research protocol. We characterize the details of ctDNA in the AH, including diagnostic and prognostic genomic biomarkers, evaluated at diagnosis and throughout treatment longitudinally with at least 12 months in follow-up. For the purpose of establishing clinical utility, we review our past publications to demonstrate the assay’s fit for purpose, focusing on the pre-analytical, analytical, and clinical validity that have empowered us to begin establishing clinical utility, prospectively, for RB [20]. 

## 2. Materials and Methods

This study was conducted under Children’s Hospital Los Angeles (CHLA) Institutional Review Board approval and adhered to the tenets of the Declaration of Helsinki. The Reporting Recommendations for Tumor Marker Prognostic Studies (REMARK) guidelines were followed (Appendix A) [21].

Patients diagnosed with RB at CHLA were included in this study if (1) written informed consent from both parents for AH sampling at the time of diagnosis and throughout therapy was attained and (2) there was at least 12 months of follow-up available. Given that the majority of intraocular and extraocular RB recurrence occurs within 1 year of diagnosis, this is an important clinical period [22,23]. RB patients who did not meet these criteria or who demonstrated any degree of anterior segment involvement on initial clinical examination that would preclude safe extraction of AH were excluded from the study (1 patient with bilateral Group E eyes was excluded due to shallow anterior chambers). There was a 0% attrition rate. Participant ages ranged from 4 to 22 months at time of diagnosis. Three participants were male, and three were female. The family histories of all participants were negative for RB; one patient (Case 44) demonstrated a germline *RB1* mutation identified via routine clinical testing of serum leukocytes. No germline *RB1* mutation was identified via clinical testing in the other cases.

The primary clinical endpoint was objective and binary: ocular salvage (the ability to treat and save the eye using standard therapies) versus enucleation (surgical removal of the eye due to persistent disease). Secondary endpoints included damage to ocular structures from the paracentesis, extraocular tumor spread, metastatic disease, and/or death.

The workflow for the AH platform has been published extensively previously [12,13,15,16,17,18,19]. Further methods are available as a Appendix A. 

## 3. Results

### 3.1. Participant Demographics and Clinical Outcomes

A total of seven eyes of six patients were included in the study as one patient (Case 44) presented with bilateral disease. No participants included in this study withdrew consent or were lost to follow-up over the study period. Case numbers remained consistent with prior studies for comparison purposes [18,19]. Demographics, diagnostic clinical features, and therapeutic courses are summarized in Table 1. Treatment courses for eye salvage were nonrandomized and decided by the treating physicians without a prior knowledge of the AH biomarkers. No patients had complications secondary to AH sampling, including infection, iris trauma, synechiae, hyphema or cataract. No child developed extraocular disease or metastatic disease throughout the follow-up period.

### 3.2. AH liquid Biopsy: Diagnostics

cfDNA was obtained and quantified in all AH samples at diagnosis (range 1.04–6.14 ng/µL, median 3.42 ng/µL, SD 1.98 ng/µL) (Figure 1). AH cfDNA at diagnosis was evaluated via low-pass whole-genome sequencing for the presence of highly-recurrent RB SCNAs including gain of 1q, 2p, 6p, loss of 13q and 16q (Figure 2). Two of the 7 eyes had no identified SCNAs at diagnosis (Cases 44_OD and 46), and 1 displayed only a focal *RB1* gene deletion (Case 44_OS). These three eyes were significantly younger in patient age at diagnosis (mean, 4.3 months; range, 4-5 months) than eyes with detectable large-scale RB SCNAs (mean, 15.8 months; range, 8-22 months; *t*-test, *p* = 0.02) as has been demonstrated previously [18].

The same AH sample at diagnosis was also evaluated for detection of *RB1* pathogenic variants. Mutational analysis of AH cfDNA identified pathogenic somatic variants with high variant allele frequency (VAF) in the 2 AH samples without SCNAs and 3 additional samples (Figure 1). VAF ranged from 66.67% to 100%, with a mean of 89.89% (Appendix A). Of the two samples without an identified *RB1* variant (Cases 33 and 48), one demonstrated focal *MYCN* amplification (Case 48); therefore, an *RB1* variant may not be expected as *MYCN* amplification has been described in the absence of *RB1* mutation [7,24,25,26]. Consistent with this hypothesis, no pathogenic *RB1* mutation was identified in the enucleated tumor tissue of Case 48. As for Case 33, oncogenesis could be due to epigenetic dysregulation, which has been suggested as a critical driver of RB tumorigenesis in the absence of an *RB1* mutation [27,28]. This would not be identified by our assay given that it covers the full length of *RB1* gene, including all exons and intron regions, but not methylation status. However, a definitive conclusion cannot be drawn based on this assay alone. 

### 3.3. AH Liquid Biopsy: Prognostics

All eyes were followed longitudinally throughout therapy for at least 12 months (median follow-up 24 months, range 18–25 months). Five eyes were cured with therapy, and two required enucleation (Cases 33 and 48) as definitive management due to persistently active intraocular disease. Both of these secondarily enucleated eyes had either a 6p gain (Case 33, with an amplitude of 1.5 ratio to the median; Figure 3a) or a *MYCN* amplification (Case 48, with an amplitude of 4.5 ratio to the median; Figure 3b) present at diagnosis, which have been previously identified as poor prognostic factors for overall globe salvage [7,18]. For the two eyes that required enucleation, SCNA profiles obtained from tumor tissue were highly concordant—92.81% for Case 33 and 96.55% for Case 48—with those obtained from the AH at diagnosis (Figure 3). Matched blood samples were also collected at time of diagnosis. All genomic profiles generated from cfDNA in the blood plasma were flat and no SCNAs were detected (Figure 3); while the presence of ctDNA in the blood plasma could not be ruled out, it is clearly below the threshold for detection of prognostic SCNAs.

### 3.4. AH Liquid Biopsy: Longitudinal Analysis

Of the four eyes with SCNAs present at diagnosis, two eyes (Cases 33 and 47) received trans-scleral injection of intravitreal melphalan (IVM) treatment for active vitreous seeding; AH samples were obtained for analysis during that treatment. cfDNA was present in significantly higher concentrations in the initial diagnostic samples (mean, 3.99 ng/µL; range, 3.42–4.56 ng/µL) than the subsequent samples obtained during IVM treatment or enucleation (mean, 0.18 ng/µL; range, 0.07–0.29 ng/µL; t-test, *p* < 0.00001). 

TFx has been shown to correlate with response to therapy both in eyes with RB as well as in other cancers [19,29]. In Case 33, TFx values remained high throughout IVM treatment, reflecting the persistently active vitreous seeding in addition to apical tumor recurrence that led to enucleation (Figure 4). In contrast, Case 47 was responsive to treatment, and the eye was ultimately salvaged. TFx was initially 79% and then decreased to <10% after systemic chemotherapy and throughout IVM treatment (Figure 5).

## 4. Discussion

Herein we present the clinical and genomic findings of the first series of RB patients to undergo prospective cfDNA analysis of the AH at the time of diagnosis and longitudinally throughout therapy. This represents an initial but essential step towards prospective validation of the clinical utility of the AH liquid biopsy platform and, most significantly, of the prognostic molecular biomarkers that were identified in earlier retrospective studies [13,16]. As with any new platform, we focused on pre-analytical, analytical, and clinical validity before establishing clinical utility [30]. In terms of pre-analytical validity, methods of specimen collection, handling, storage, and processing have been standardized [12,13,15,16,17,18,19]. In previously published evaluation of >100 samples [16], less than 5% of AH samples were removed for quality control due to poor reads alignment ratio. We previously reported concentrations of cfDNA isolated from AH samples (median 0.2 ng/µL, range 0.084–56 ng/µL) [13] but obtained even higher concentrations (median 3.42 ng/µL, range 1.04–6.14 ng/µL) in the diagnostic samples described here.

The analytical validity of the AH liquid biopsy platform to accurately and reliably detect SCNAs and *RB1* pathogenic mutations is well documented. A commonly accepted measure of assessing analytical validity in ctDNA studies is concordance of genomic findings with that from matched tumor tissue [31]. We previously reported a mean concordance of 94.1% (range 84.3–100%) between genomic profiles generated from AH ctDNA samples and corresponding tumor tissue (13 pairs) [13]. Herein, we demonstrated 92.81% and 96.55% concordance between AH and tumor tissue for the two enucleated eyes. 

Finally, in terms of clinical validity, using this platform we demonstrated that 6p gain in the AH ctDNA was associated with 10-fold increased odds of enucleation at our treatment center; this prognostic biomarker was the only independent predictor of ocular salvage and has been evaluated in 116 samples from 50 RB eyes with greater than 24 months of clinical follow-up [13,16]. Our platform is also capable of identifying *MYCN* amplification, a known biomarker of aggressive RB [7,24,25,26]. Based on this data [16], we have established a molecular signature based on presence of either 6p gain with an amplitude of ≥1.5 ratio to the median (which corresponds to at least one full copy gain) or *MYCN* amplification. This signature is prognostic for a 16.5-fold increased likelihood of treatment failure requiring enucleation. This model is 74% sensitive and 85% specific for the prediction of ocular salvage; the positive and negative predictive values are 81% and 79%, respectively. Notably, enucleation is a terminal and clearly defined end point, as the RB eye is either saved or not at a given point in time. Given that all current predictions regarding the likelihood of therapeutic success rely upon clinical features only, the benefit of an objective biomarker in the management of these young patients with RB is clear and compelling. However, given that treatment can be rather heterogenous across RB centers, this prognostic signature has only been evaluated at our institution; external validation is needed prior to broad clinical implementation.

Taken together with pre-analytical, analytical, and clinical validity, this study provides the first prospective evidence for establishing clinical utility. Here, we demonstrate that our comprehensive RB workflow provides clinically relevant tumoral information from a single 100µL diagnostic sample of AH. In terms of diagnostic value, this workflow can identify pathogenic *RB1* variants that induce tumorigenesis in both heritable and sporadic cases, whereas commonly utilized blood-based biopsies are only capable of identifying germline, heritable mutations [17]. However, it should be noted that the detection of somatic *RB1* mutations in the plasma has recently been described and is an active area of research [32]. Furthermore, our RB workflow detects prognostic molecular biomarkers, specifically 6p gain and *MYCN* amplification, that objectively predict the likelihood of therapeutic response and eye salvage [16,19]. Longitudinal analysis of the AH is also impactful; TFx trends can help monitor for therapeutic response as increases in TFx portend disease recurrence [19]. Altogether, this comprehensive, sensitive AH workflow paves the way for a future precision oncology model of RB management in which treatment decisions are made based on the specific genomic profile of each patient and each eye. 

From a clinical perspective, this initial prospective study demonstrates that analysis of the AH taken at the time of diagnosis is feasible, safe, and there is clear clinical utility. Years of evidence support an extremely low risk of complications from AH aspiration in the setting of IVM injections, with no patients displaying tumor spread or seeding into the needle tracts or beyond [12,13,14,15,16,17,18,19,33]. With demonstrated safety and utility, prospective analysis of treatment-naïve eyes prospectively starting at diagnosis could be initiated. As with previous retrospective studies, there was no evidence of either acute or chronic complications at a minimum of 12 months follow-up, and no evidence of extraocular tumor spread. As a safety measure, diagnostic paracentesis is only performed if the anterior chamber is clear and formed, the pressure is normal (<22 mmHg), and there is no evidence of anterior tumor involvement. These safety criteria resulted in the exclusion of one patient in this yearlong study with shallow anterior chambers in the setting of bilaterally advanced disease. Many Group E eyes may similarly have shallow chambers preventing safe AH extraction. Often, primary enucleation is performed based on clinical findings. However, if enucleation is not done, then AH could likely be safely extracted after the first round of systemic or intra-arterial chemotherapy when the tumor shrinks, and the chamber deepens. Regardless of this early safety data, we emphasize that there is no approved clinical protocol for routinely performing an AH liquid biopsy. Multi-center prospective research studies are required before AH analysis at diagnosis and throughout therapy can become a standard aspect of RB patient care. 

It has been hypothesized that the source of AH ctDNA in RB is primarily from necrosis of posterior segment tumor cells, a process which could be affected by both systemic and local treatment [12,34]. Thus, it was unknown whether tumors in treatment-naïve eyes would have substantial intratumoral necrosis to shed detectable tumor-derived cfDNA into the AH. Furthermore, the majority of eyes in previous AH liquid biopsy studies had advanced disease (Groups D and E) with active vitreous seeding; therefore, it was also unclear whether eyes with smaller tumor burden and/or the absence of seeding would shed measurable amounts of tumor DNA into the AH [12,13]. This study included two eyes with less advanced disease (Group B and C) and three with an absence of vitreous seeding, yet detectable levels of tumor-derived cfDNA were identified in every diagnostic sample of AH based either on the identification of SCNAs, pathogenic *RB1* SNVs, or both. In fact, the highest levels of cfDNA were seen in diagnostic samples, with concentrations falling significantly after treatment was initiated. Although the identification of SCNAs is not expected in all RB eyes [4,35,36], we hope to improve our detection rate for somatic *RB1* mutations with refinements to our assay, specifically with the inclusion of methylation analysis. Currently, the AH liquid biopsy remains the only described platform for RB that identifies ctDNA via two mechanisms: the presence of SCNAs and *RB1* SNVs. A benefit of an AH based platform is the ability to detect eye specific prognostic SCNAs. Detection of SCNAs in the blood is limited by lower ctDNA TFx and the inability to correlate with each eye in the 40% of patients with bilateral disease [15].

Other liquid biopsy platforms have been described for retinoblastoma. The detection of ctDNA based on *RB1* SNVs in the AH has been described by Gerrish et al. [14] and Kothari et al. first described *RB1* SNVs detected from the plasma of patients with advanced intraocular disease requiring enucleation [32]. To the authors’ knowledge, blood based liquid biopsy platforms are actively being investigated at Memorial Sloan Kettering Cancer Center as well as Institut Curie, while development of AH platforms is ongoing at Birmingham Women’s, Children’s NHS Trust, and CHLA. 

Beyond substantiating the presence of tumor-derived cfDNA, identification of somatic *RB1* mutations in such a liquid biopsy has clear clinical utility. First, it facilitates tumor-directed mutational analyses to improve sensitivity of peripheral blood testing for germline *RB1* mutations [17]. This is particularly relevant for identifying cases of mosaicism, which is important for accurate family genetic counseling. Second, the novel ability to characterize tumor-specific somatic mutations in vivo, at diagnosis, creates opportunity for larger, prospective studies investigating the prognostic value of specific somatic *RB1* alterations or other pathogenic mutations, such as *BCOR* or *CREBBP* [24]. This was previously impossible due to the inability to identify somatic pathogenic variants in the blood and the lack of access to tumor DNA prior to enucleation. A third application is the development of an *RB1*-VAF based pipeline for TFx estimation. TFx has been shown to correlate with RB therapeutic response [19]. The commonly used software (ichorCNA) relies on SCNAs to estimate TFx [37]. However, this approach is not applicable for the approximately 30% of RB tumors that contain no large-scale SCNAs [19] (e.g., cases 44_OD, 44_OS, and 56). This constraint is pertinent for RB patients diagnosed at younger ages, as these tumors are less likely to demonstrate SCNAs [18]. Because RB tumors (with few exceptions [7,24,25,26]) harbor somatic *RB1* mutations, an *RB1*-based VAF pipeline may provide a better measurement of tumor burden and therapeutic response [19]. 

An important implication of this study is the possibility of using tumor-derived, eye-specific cfDNA molecular biomarkers to provide prognostic information for *each eye* at the time of diagnosis. As 40% of children with RB have tumors in both eyes, it is critical that any prognostic biomarkers for the likelihood of ocular salvage be specific to the eye. Diagnostic biopsy in combination with genetic analysis has proven useful for other malignancies (including ocular cancers) and facilitates a patient-specific approach to cancer management [38,39,40,41,42]. Our study is the first to identify these prognostic biomarkers—specifically 6p gain with an amplitude of ≥1.5 ratio to the median (Case 33) and *MYCN* amplification (Case 48)—present in non-enucleated eyes, at the time of diagnosis. Additionally, these two cases were the only eyes that required enucleation throughout our study period. In concert with retrospective analysis of more than 100 AH samples in 50 RB eyes, this lends initial prospective evidence that the presence of either of these biomarkers may portend a poor prognosis for saving the eye. However, we emphasize that our current conclusions are limited due to small sample size and that further prospective and external validation of the AH liquid biopsy platform are needed.

## 5. Conclusions

This initial prospective study demonstrates the clinical utility of AH liquid biopsy for RB at diagnosis and longitudinally during therapy for its: (1) diagnostic value, (2) prognostic significance, and (3) potential for future application to a precision oncology model to direct personalized management of RB. This novel approach finally allows for the identification of tumor-derived molecular biomarkers in RB eyes without invasive tumor biopsy or enucleation. The AH liquid biopsy opens the door to apply decades of knowledge about RB genomics in an impactful clinical application, and to better understand intratumoral dynamics throughout therapy. The overall goal of implementation of an AH liquid biopsy is to enhance the care of patients with RB based on objective diagnostic and prognostic molecular biomarkers; this publication is the first step towards establishing this clinical utility. While further prospective research across multiple treatment centers is necessary before a companion diagnostics approach can be implemented, a new era of personalized, precision oncology RB management is on the horizon.

## 6. Patents

Drs. Berry, Xu, and Hicks have filed a provisional patent application entitled Aqueous Humor Cell Free DNA for Diagnostic and Prognostic Evaluation of Ophthalmic Disease.

## Figures and Tables

**Figure 1 cancers-13-01282-f001:**
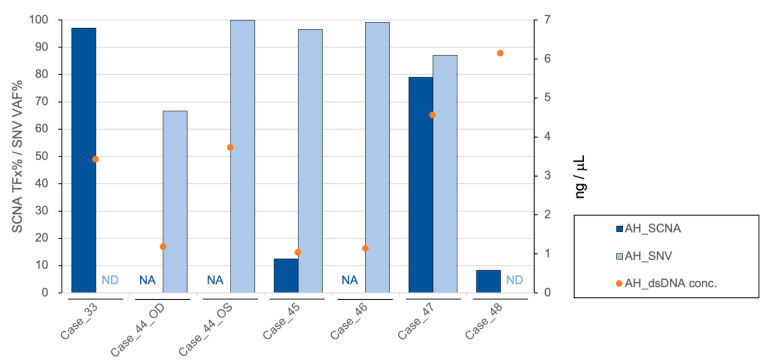
Diagnostic information from AH liquid biopsy with graph showing the tumor fraction (TFx) of SCNA (**dark blue bar**), variant allele frequency (VAF) of *RB1* SNV (**light blue bar**), and cfDNA quantification (**orange dots**) for each case’s AH sample at time of diagnosis. NA: not available; ND: not detected.

**Figure 2 cancers-13-01282-f002:**
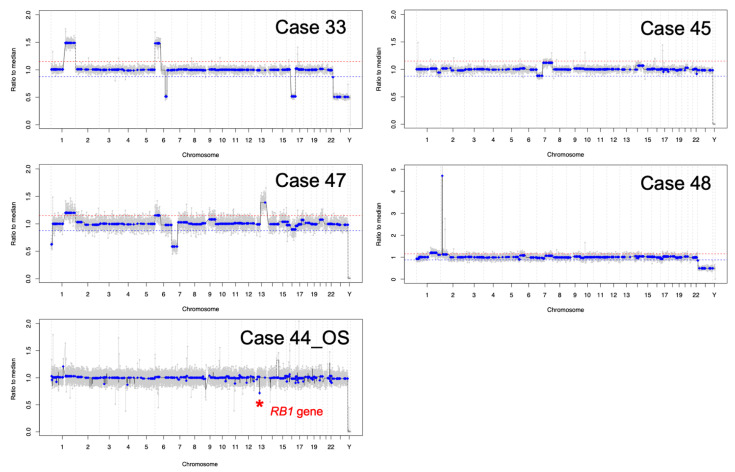
Genomic profiles for the 4 eyes with detectable large-scale SCNAs (cases 33, 45, 47, 48) and the 1 eye with a focal *RB1* gene deletion (Case 44 OS).

**Figure 3 cancers-13-01282-f003:**
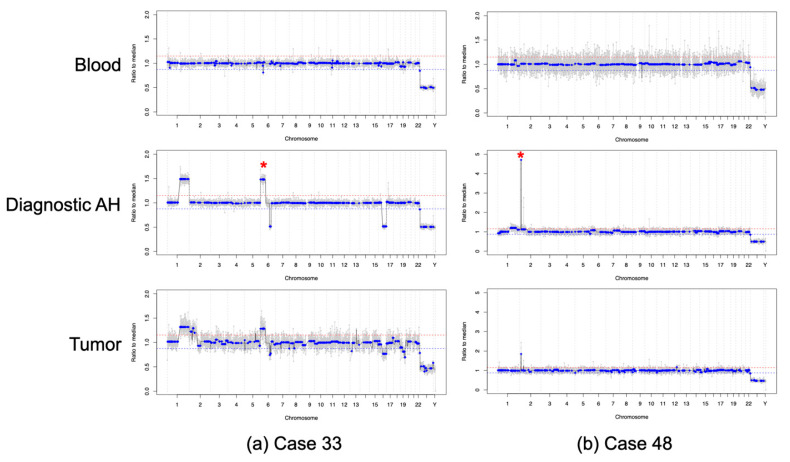
Prognostic information from AH liquid biopsy. Genomic profiles from the blood, AH sample at time of diagnosis, and tumor tissue at time of enucleation for (**a**) Case 33, asterisk showing 6p gain with an amplitude of 1.5 ratio to the median and (**b**) Case 48, asterisk showing *MYCN* amplification. Genomic profiles from the blood were without SCNAs, compared to the genomic profiles from the AH at time of diagnosis, which were highly concordant with those obtained from the tumor tissue. Due to admixing with normal retinal tissue, SCNAs from tumor tissue may show lower amplitude compared to AH due to diluted TFx.

**Figure 4 cancers-13-01282-f004:**
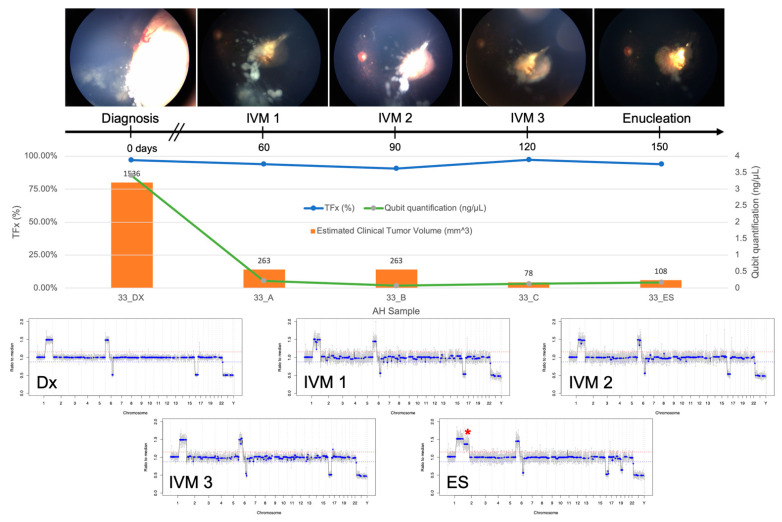
Longitudinal information from AH liquid biopsy for Case 33. Fundus photos, tumor fraction (TFx), cfDNA quantification, estimated clinical tumor volume from B-scan measurements, and genomic profiles for each clinical timepoint at which an AH sample was evaluated. This eye was non-responsive to treatment, and ultimately required secondary enucleation (ES) due to an apical tumor recurrence with persistently active seeds. A dramatic decrease in cfDNA quantity was observed over time, consistent with the decrease in the main tumor volume. This indicates that the total number of tumor cells that were actively shedding cfDNA into the AH decreased. However, TFx remained at a high level throughout treatment, reflecting the persistent seeding that still contributes to tumor-derived cfDNA in the AH. Without much physiological cfDNA to dilute the tumor-derived cfDNA, the TFx remained high. Genomic profiles consistently demonstrated the same 3 SCNAs that were present at diagnosis, as well as a new large-scale 2p gain (*) and 19q loss in the AH of the enucleated eye, suggesting clonal evolution at the time of recurrence.

**Figure 5 cancers-13-01282-f005:**
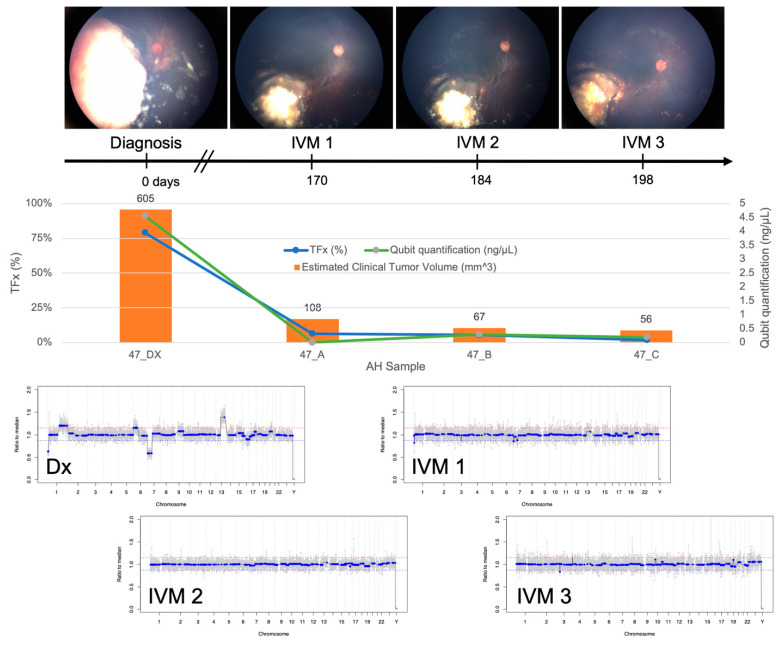
Longitudinal information from AH liquid biopsy for Case 47. Fundus photos, tumor fraction (TFx), cfDNA quantification, estimated clinical tumor volume from B-scan measurements, and genomic profiles for each clinical timepoint at which an AH sample was evaluated. This eye was responsive to treatment and remained salvaged at 19 months of follow-up. A decrease in cfDNA quantity was observed over time, consistent with the decrease in the main tumor volume. TFx also decreased, reflecting resolution of seeding. Genomic profiles completely normalized over treatment, as would be seen with clinical regression of disease.

**Table 1 cancers-13-01282-t001:** Clinical demographics, diagnostic clinical features, and therapeutic courses for each study participant.

Case	Demographics		Diagnostic Characteristics					Treatment Course						
	Sex	Age at Dx (mos)	IIRC Group	TNM Stage	Laterality	Seeding Type	Blood RB1 Mutation	Initial Tx	Required IVM?	Required ENUC?	Reason For ENUC	Time to ENUC after Dx (mos)	F/u from Dx (mos)	F/u from Initial Tx Completion (mos)
33	M	22	D	cT2b	U	dust	negative	IAC	yes	yes	persist	6	24	18
44_OD	F	4	B	cT1b	B	none	c.1666C>T	CEV	yes	no	NA	NA	25	19
44_OS	F	4	D	cT2b	B	dust	c.1666C>T	CEV	yes	no	NA	NA	25	19
45	F	8	D	cT2b	U	no vitreous,+ subretinal	negative	CEV	no	no	NA	NA	24	18
46	M	5	C	cT2a	U	none	negative	CEV	no	no	NA	NA	24	18
47	F	15	D	cT2b	U	sphere	negative	CEV	yes	no	NA	NA	19	13
48	M	18	D	cT2b	U	cloud	negative	IAC	no	yes	persist	1	18	18

Abbreviations: OD, right eye; OS, left eye; M, male; F, female; Dx, diagnosis; IIRC, International Intraocular Retinoblastoma Classification; Tx, treatment; IVM, intravitreal melphalan; ENUC, enucleation; F/u, follow up.

## Data Availability

The data presented in this study are available on request from the corresponding author. Due to NIH funding, the data are available to other researchers via NIH GDS/dbGAP controlled databases and are also available to the public upon request from the corresponding author.

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
