# Peer review of "Establishing the Clinical Utility of ctDNA Analysis for Diagnosis, Prognosis, and Treatment Monitoring of Retinoblastoma: The Aqueous Humor Liquid Biopsy"

_cancers, 2021, doi:10.3390/cancers13061282_

Round 1

Reviewer 1 Report

The most concerning aspect of this study is the small volume. They include only 7 eyes in 6 patients. It is difficult to draw meaningful conclusions from such a small sample size. The authors make hefty statements based on an underpowered cohort. For instance, they write, “our study is the first to identify these prognostic biomarkers- specifically 6p gain and MYCN amplification- present at the time of diagnosis”. They have only one eye with 6p gain and one eye with MYCN amplification. The authors may want to be cautious in overstating their conclusions based on an “n” of only one. More meaningful conclusion are supported by robust data on higher volume cohorts, and their paper would benefit from clearly stating this.

The authors write extensively on their discovery that 6p gain is “prognostic for a 16.5-fold increased likelihood of treatment failure requiring enucleation”. However, as the authors well know, treatment approaches and modalities are not uniform across different retinoblastoma centers. An eye destined for enucleation at one center may be salvaged at a different center. It is advisable for the authors to point out that 6p gain is a prognostic feature for enucleation for eyes treated at their center, and to clarify that this prognostic feature may not hold true at other centers that treat retinoblastoma. As it now stands, their statements are misleading and warrant clarification.

It is very surprising that two eyes (29%) did not have identifiable somatic chromosomal alterations and another two eyes (29%) did not have identifiable RB1 variants. Their assay did not identify expected alterations in almost one third of the cases. This suggests a concerns for a high false-negative detection rate. Please explain and discuss.

The authors state, “all genomic profiles generated from cfDNA in the blood plasma were flat and no SCNAs were detected”. This is in contrast to other plasma-based platforms that have been described in the literature. Please provide an explanation for why their assay could not identify genomic profiles in plasma. Would this suggest their assay possesses a less sensitive detection rate and therefore at higher risk for false-negatives?

The authors write, “in terms of diagnostic value, this workflow can identify pathogenic RB1 variants that induce tumorigenesis in both heritable and sporadic cases, whereas commonly utilized blood-based biopsies are only capable of identifying germline, heritable mutations”. This statement is true, but outdated. The authors may be aware that cell-free DNA has been identified in the plasma of patients with retinoblastoma: allowing for the identification of both somatic and germline mutations. It is advisable for the authors to update their discussion by including references to these plasma-based cell-DNA platforms.

In the introduction and discussion, it would be helpful for the authors to describe the other cell-free DNA platforms that have been described for retinoblastoma. This will make for a more well-rounded discussion and informed paper. Please extrapolate further on the work described in reference number 14, in addition to plasma-based platforms.

The authors describe a patient that was excluded from the study due to advanced disease, shallow chamber and risks of aqueous humor acquisition.

Please make mention of this exclusion criteria in the discussion. In the fifth paragraph in the discussion, the authors describe the safety profile of their procedure, however, it would be beneficial to point out that some cases would be deemed unsafe to enter with a needle.

It is unclear how the current approach of AH sampling and genetic information provide additive information beyond simple indirect ophthalmoscopy and interpretation of fundus details. Examination and correct interpretation will inform the clinician regarding activity of the tumor. What does the current platform provide beyond that?

The authors have included patients with both unilateral and bilateral disease. It is unclear if their assay can distinguish between germline and somatic alterations? Please explain. This would be particularly pertinent to bilateral patients: how do the authors know if the alterations they detect in one eye are derived from somatic mutations in that eye or reflective of germline alterations?

Reviewer 2 Report

Analysis of tumor DNA present in aqueous humor (AH) obtained from eyes with active retinoblastoma is a current hot topic of translational research in retinoblastoma. This manuscript presents data from the first year of a prospective study that monitored 6 RB patients. In total, AH was obtained from 7 eyes at several time points including time points prior to therapy.

The data presented in this manuscript will be to great interest of many readers who work in the retinoblastoma field. The results justify the hope that studies on a larger set of patients may provide sufficient evidence that patients may benefit from analysis of tumor DNA in AH.

The data presented here do not support some of the enthusiastic conclusions that the authors draw (see below for details). I can understand the reasons for such hyperbole. However, a more restrained interpretation of findings will be exciting enough to attract the attention of readers.

Some points to consider:

line 34ff: "we demonstrate that molecular profiling of the AH at diagnosis and longitudinally throughout therapy has significant clinical utility for diagnosis, prognosis, and monitoring of treatment response." AND
line 330ff: "This initial prospective study demonstrates the clinical utility of AH liquid biopsy for RB at diagnosis and longitudinally during therapy for its (1) diagnostic value, (2) prognostic significance, ..."
REMARK: Even if Rb was a homogeneous disease with respect to presentation, management, prognosis, etc (and it is NOT), data from 7 eyes/6 patients are hardly sufficient to support the claims, esp. "significant clinical utility". 

line 38: "somatic chromosomal alterations"
REMARK: The assay will detect large-scale copy number variation/alteration only.

line 132ff: "while all others were without a germline RB1 mutation"
MORE PRECISE: "no germline mutation identified" (BTW, what was the scope of analysis of constitutional DNA?)

line 101ff: Data on laterality, family history better in "2. Materials and Methods"

line 162ff: "Of the two samples without an identified RB1 variant (Cases 33 and 48), one demonstrated focal MYCN amplification (Case 48); therefore, an RB1 mutation may not be expected [7].
REMARK: Findings and conclusions in citation 7 have been invalidated. Therefore, the statement "therefore, an RB1 mutation may not be expected" reflects a state of knowlege that is no longer valid.

line 164ff: "As for Case 33, oncogenesis could be due to epigenetic dysregulation, which has been suggested as a critical driver of RB tumorigenesis in the absence of an RB1 mutation [24, 25]"
REMARK: The assays used for identification of RB1 variants cover only a part of the diverse inactivating alterations. Promoter hypermethylation is only part of this part. The authors should state what part of the spectrum of pathogenic variants may be detected.

line 185: "it is clearly below the threshold for detection of prognostic SCNAs"
REMARK: Threshold of detection is a characteristic of a method.
It is clearly below the threshold of detection of the methods used here. But other methods may do.

Figure 3 (and manuscript text): Why is the quantitative fidelity of SCNA analysis low (lower gains/losses in DNA from tumor)?

Figure 4: Sample ES also seems to show chr 19 loss (in addition to 2p gain).

line 239ff: "The analytical validity of the AH liquid biopsy platform to accurately and reliably detect ctDNA"
REMARK: Analytical validity is about intended use (see international definition). Intended use presented here is CN profile and SNVs in RB1. Not "detect ctDNA".

line 258ff: "current predictions regarding the likelihood of therapeutic success rely upon clinical features only, the benefit of an objective biomarker in the management of these young patients with RB is clear and compelling."
QUESTION: Is this a realistic scenario?:
If, according to the results of the biomarker test, therapeutic success is not likely then enucleation would be performed just after initial diagnosis instead of eye-preserving therapies even if these might be possible as of current knowlege?

line 266ff: "whereas commonly utilized blood-based biopsies are only capable of identifying germline, heritable mutations [17]."
REMARK: This is not correct, for several reasons. One is somatic mosaicism, another is ctDNA in plasma.

line 300ff: "First, it facilitates tumor-directed mutational analyses to improve sensitivity of peripheral blood testing for germline RB1 mutations [17]."
REMARK: I do not understand what the authors would like to state here -- possibly other readers will be clueless as well.

line 324ff: "Additionally, these two cases were the only eyes that required enucleation throughout our study period, lending further prospective evidence that the presence of either of these biomarkers portends a poor prognosis for saving the eye."
REMARK: Two of seven observations. Enough for hope but not for such a strong statement.

Round 2

Reviewer 1 Report

The authors have partially addressed the reviewers prior comments. However, there are a few remaining/new comments to be addressed.

  1. The authors state that their primary clinical endpoint was ocular salvage versus enucleation. They also state that the majority of recurrence occurs within a year of diagnosis; however, recurrences typically occur with a year of treatment completion (the basis of this information is on eyes that were enucleated following diagnosis and thusly diagnosis/treatment dates coincide, but for eyes that are salvages, these dates can be disparate). They write their follow up was “at least 12 months”. Please define the follow up period: is this follow-up from diagnosis, or follow-up from treatment completion? If the later, please provide the follow-up from treatment completion. This has important implications regarding the authors primary clinical endpoint and should be clearly stated.
  1. The authors identified MYCN amplification in an eye without identifying an RB1 variant (case 48). They speculate this eye maybe wildtype RB1 as has been demonstrated by the work of Rushlow. This eye was enucleated. To be complete, please provide the genetic information on the enucleated eye: was an RB1 variant detected?
  2. Please state whether the assay can identify large deletions of RB1, and LOH chromosome 13.
  3. Please revise the updated sentence so that it is clearer for the reader (move the “at our treatment center” addition). The current sentence is: “Finally, in terms of clinical validity, using this platform we demonstrated that 6p 244 gain in the AH ctDNA was associated with 10-fold increased odds of enucleation; this 245 prognostic biomarker was the only independent predictor of ocular salvage and has been 246 evaluated in 116 samples from 50 RB eyes at our treatment center with greater than 24 247 months of clinical follow-up[13, 16].” Please revise to: “Finally, in terms of clinical validity, using this platform we demonstrated that 6p 244 gain in the AH ctDNA was associated with 10-fold increased odds of enucleation at our treatment center; this 245 prognostic biomarker was the only independent predictor of ocular salvage and has been 246 evaluated in 116 samples from 50 RB eyes with greater than 24 247 months of clinical follow-up[13, 16].”
  4. The significance of MYCN amplification is cited by these authors. However, they use a reference for eyes that have MYCN amplification in the context of wildtype RB1. Please provide a reference for the association of MYCN amplification to aggressive features in the context of RB1 variant.

Reviewer 2 Report

Thank you for considering my remarks.

Author Response

Thank you for your review and opportunities to improve our manuscript.

Round 3

Reviewer 1 Report

The authors have sufficiently addressed the concerns of the reviewer.